# Feasibility of serial measurement of nitrite for pharmacodynamic monitoring and precision prescribing in urinary tract infections

Ellen V. Stadler [1,2,3] ✉, Alison Holmes [1,2,3,4,5], Danny O'Hare[1,6], Mark Sutton[7,8], Colin Brown[9] & Timothy M. Rawson [1,3,4]

## Abstract

**Background** The ability to monitor host- and bacteria-specific biomarkers along with antimicrobial drug concentration at the site of infection offers potential for individualised approaches to antimicrobial therapy. Although urine collection is straightforward and directly linked to the infection site, the assessment of urinary tract infection (UTI) biomarkers during infection has not been extensively explored. The aim of this study is to evaluate the potential of monitoring urinary nitrite levels as a biomarker for antimicrobial pharmacodynamics in UTI treatment.

**Methods** Resistant and susceptible *E. coli* strains were cultured in oxygen-free artificial urine, with amoxicillin added after 15 h. Colony-forming unit (CFU) counts, nitrite, and creatinine levels were measured at 5 timepoints over 66 h. Urine samples from 25 UTI patients and 25 non-UTI controls were analysed for bacterial growth, nitrite, and creatinine. Spearman rank correlation and Mann-Whitney U-tests were used for statistical analysis.

**Results** Our in-vitro model demonstrates that measuring the bacteria-specific urinary biomarker nitrite during *E. coli* growth in artificial urine can effectively be applied to assess antimicrobial pharmacodynamics over the course of UTI treatment. In an in-vitro UTI model, nitrite concentration can differentiate between resistant and susceptible *E. coli* strains and correlates with CFU counts. Analysis of 25 clinical UTI samples is consistent with these findings, showing correlations between nitrite levels and CFU counts.

**Conclusions** Here we show that nitrite generation by *E. coli* may have clinical relevance as a biomarker for infection progression and antimicrobial treatment outcomes, offering a valuable tool for monitoring the pharmacodynamic responses to antimicrobial therapy in UTIs.

## Plain language summary

Urinary tract infections are common bacterial infections, and biomarkers like nitrite are often used to support their diagnosis. This study explores monitoring nitrite to assess urinary tract infection (UTI) progression and response to treatment. We show that amoxicillin-resistant bacteria continue producing nitrite during treatment, while susceptible bacteria stop producing the biomarker when their growth is effectively inhibited. We also show that as bacterial levels increase in artificial urine and patient urine, nitrite concentrations increase too. We show that measuring nitrite levels over time during a UTI could help doctors choose the best antibiotic and adjust treatment for each patient.

Antimicrobial resistance is a global health threat[1], and urinary tract infection (UTI) is a common indication for antimicrobial therapy[2]. Although fixed antimicrobial dosing is often applied, this one-size-fits-all approach is not optimal. Both pharmacokinetics (PK), describing how a drug is metabolised and cleared in a patient, and pharmacodynamics (PD), the effect of the drug on infecting organism, vary significantly between patients and within the same patient over time[1].

Potential approaches to optimise treatment whilst minimising the occurrence of drug-resistance include using diagnostic tools to assess antimicrobial PD based on host response[1]. Current PK-PD targets for organisms are typically defined by serum antimicrobial PK and in-vitro minimum inhibitory concentration (MIC). These measurements do not necessarily reflect the antimicrobial drug's effect in urine at the site of infection[3]. Monitoring bacteria specific urinary biomarkers like nitrite alongside host-response biomarkers directly in the urine could provide a more relevant measure of antimicrobial efficacy and bacterial response in the infection site.

UTI diagnosis is commonly supported by detecting nitrite with dipsticks[4–6]. Nitrite levels are rarely reported quantitatively in UTI diagnosis

and nitrite has not been explored as a measure of organism response to antimicrobial therapy longitudinally in patients. Ottiger et al. found that urinary leucocyte levels in women with UTI decreased after effective treatment, but not when treatment failed[7], highlighting the potential of monitoring host-response biomarkers to assess treatment efficacy. Li et al. observed that after successful treatment of UTI, nitrite concentration is decreased in a rat model[8]. Further exploration of urinary biomarkers to measure antimicrobial pharmacodynamics and treatment success is therefore merited.

In this study, we investigate the viability of quantitatively monitoring urinary nitrite concentration to assess the exposure response to antimicrobial treatment in UTIs. In an in-vitro model, we show that nitrite levels correlate with bacterial growth of *E.coli* and differentiate between resistant and susceptible *E.coli* strains when treated with antimicrobial drug. These findings are consistent with clinical UTI samples, where we observed a correlation between CFU and nitrite concentration. This study suggests nitrite is a valuable biomarker for monitoring infection progression and antimicrobial treatment outcomes in UTIs.

## Methods
### Bacterial strain and growth media
Artificial urine (AU) with a physiological nitrate concentration of 20 mM was prepared following the procedures outlined in ref. 9 and was sterilised using a syringe filter (Corning, nylon membrane, diam. 25 mm and pore size 0.2 μm). AU with a concentration of $1.5 \times 10^8$ CFU/mL of *E.coli* strain 25922 or strain 35218 was achieved by comparing absorbance to a 0.5 McFarland standard. This solution was diluted to a final concentration of $1.5 \times 10^3$ CFU/mL, aligning with the typical bacterial count observed in uncomplicated UTI[10].

### Growth conditions and antimicrobial killing
*E.coli* cultures were cultivated in 1 mL volumes within 96-deep well plates with non-oxygen permeable lids to create anaerobic conditions. All samples were incubated in a single batch, using two 96-deep well plates which were filled with triplicates of *E.coli* strain 35218 and 25922 for each time point with a known starting concentration of $1.5 \times 10^3$ CFU /mL. 164 mg/mL sterilised amoxicillin, a concentration high enough to inhibit the growth of high bacterial count cultures, was introduced to samples after 15 h of growth. To minimise exposure to oxygen, the non-oxygen permeable lid was only removed from sample triplicates that were analysed at the time-point of interest.

At each time-point, samples of both bacterial strains were removed from the 96-well plate and transferred to Eppendorf tubes. A range of dilutions were immediately plated on Mueller-Hinton agar plates for CFU determination. The Eppendorf tubes containing the samples were stored at −18 °C to measure nitrite concentrations within a 72 h timeframe. All nitrite measurements were performed simultaneously at the end of the in-vitro study, alongside a series of AU samples spiked with known nitrite concentrations to generate a calibration curve. All experiments were conducted in triplicates.

### Growth determination
For quantifying CFU, the specimen was diluted ($10^{-2}, 10^{-3}, 10^{-4}, 10^{-5}, 10^{-6}$) and subsequently plated on Mueller-Hinton agar plates. Following 24 h of incubation, CFUs were counted manually.

### Nitrite and creatinine measurement
Nitrite was assessed using the Griess method[11] with a detection range up to 100 μM and a limit of detection (LOD), defined according to the IUPAC as 3.3 times standard deviation of the intercept divided by the slope of the calibration[12], of 5.9 μM. Briefly, 1 mg/mL N-(1-naphthyl)ethylenediamine dihydrochloride was mixed with 10 mg/ mL sulfanilic acid in 5% (w/w) orthophosphoric acid to form the Griess reagent.

AU samples were centrifuged for 5 min at 2000 rpm. 40 μL of the supernatant was mixed with 10 μL of Griess reagent and shaken for 15 min before reading the absorbance at 548 nm. Creatinine was measured using the Jaffe method as described in ref. 13. 100 μM of a working reagent consisting of equal volumes of 17.5 mmol/L picric acid and 0.29 mol/L sodium hydroxide were mixed with 10 μM of the sample of interest. Absorbance was read in 96-well plates in a FLUOstar Omega plate reader after 90 s of incubation. The assay has an upper detection limit of 20 μM and a LOD of 2.18 μM according the IUPAC definition[12].

### Urine collection of patients with and without UTI
Urine samples of twenty-five female patients with uncomplicated UTI (confirmed growth of *E. coli* $> 10^5$ CFU/mL, white cell count (WCC) $> 50$–100, no epithelial cells) and 25 female patients without UTI (no suspected infection, no bacterial growth, WCC $< 50$, no epithelial cells) were stored in boric acid at 4 °C until further analysis. Boric acid preserves bacterial counts and has no known effect on nitrite detection using dipstick tests[14]. At collection, the samples were routinely analysed for growth of organisms, white cell count and epithelial cells at Northwest London Pathology, Imperial College Healthcare NHS Trust. Bacterial growth determination, nitrite and creatinine measurement was conducted 4 days after collection at Hammersmith Hospital. Regional Ethics Committee approval was obtained prior to the start of study and is summarised in IRAS Project ID 162013, COREC Application 06/Q0406/20: *The detection of microbial products and effects on patients in infection*. The study protocol allows use of clinical samples routinely submitted to a Diagnostic laboratory as part of patient care, and where written consent is not normally obtained.

### Statistical analysis
Statistical analyses were performed using Spearman rank correlation and Mann-Whitney U-tests. Spearman rank correlation was used to assess the monotonic relationship between nitrite levels and CFU counts for each strain under different conditions and in UTI patients. The Spearman correlation S was calculated, and a *p*-value of less than 0.05 was considered statistically significant. Spearman correlation was used instead of a Pearson correlation as a normal distribution of values was not provided.

Mann-Whitney U-test was used to compare nitrite concentrations and CFU counts between treated and untreated groups for each strain after addition of antimicrobial drug, as well as to evaluate differences in nitrite levels between UTI and non-UTI patient groups. Statistical significance was defined as $p < 0.05$ for all tests. Mann-Whitney U-test is used to determine if there is a significant difference between two groups, without assuming any distribution of values.

Statistical analyses were conducted using Python with the following packages: pandas, numpy, and scipy for statistical tests, alongside matplotlib and seaborn for data visualization.

### Reporting summary
Further information on research design is available in the Nature Portfolio Reporting Summary linked to this article.

## Results
### Nitrite generation of resistant and susceptible *E. coli* strains in artificial urine
Addition of amoxicillin after 15 h incubation inhibits the growth of the susceptible *E. coli* 25922. Addition of amoxicillin to the resistant strain, *E. coli* 35218, does not affect growth with continued proliferation until reaching a growth plateau (Fig. 1a, b). Nitrite production mirrors the CFU (colony-forming unit) counts, with Spearman rank correlation analysis revealing significant relationship between nitrite levels and CFU counts in all strains namely the susceptible strain, susceptible strain with drug, resistant strain, and the resistant strain with drug (Spearman correlation coefficient of 0.82 and *p*-value < 0.001, 0.62 and 0.014, and 0.86 and <0.0001 and 0.79 and <0.001, respectively) (Fig. 2a–d). The nitrite concentration and CFU count after addition of antimicrobial drug is significantly lower in the susceptible strain with drug compared to

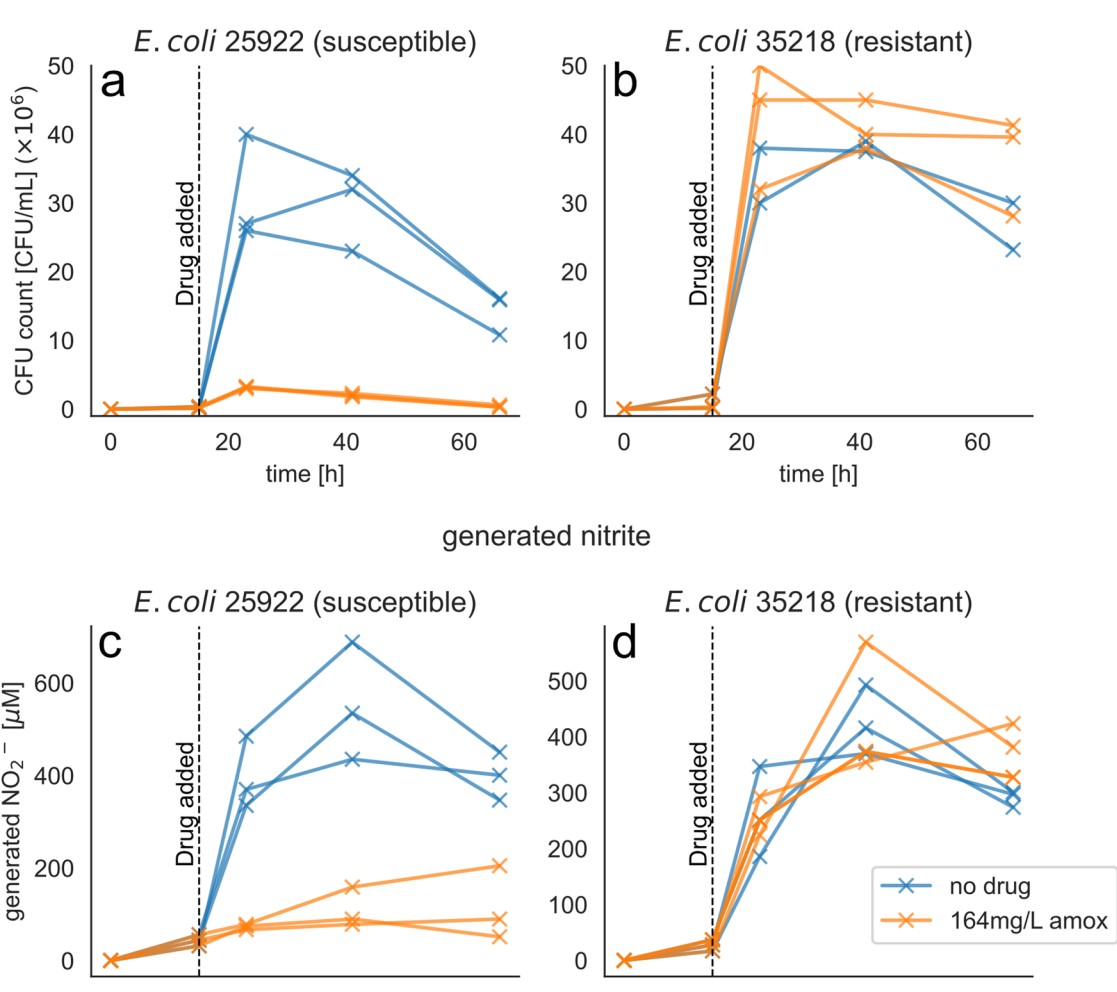

**Fig. 1 | Bacterial growth and generated nitrite of susceptible and resistant *E. coli* in artificial urine. a** 164 mg/L amoxicillin inhibits the growth of *E. coli* 25922. **b** *E. coli* strain 35218 is resistant to amoxicillin and growth is not inhibited. **c** Total produced nitrite is only increasing sparsely when susceptible *E. coli* is subjected to amoxicillin. **d** The addition of antimicrobial drug has no effect on nitrite production in a resistant *E. coli* strain.

without drug (Fig. 2e, f, Mann-Whitney U-Test: U = 81.0, *p*-value < 0.001 for nitrite and U = 81.0 and *p* < 0.001 for CFU count), whereas there is no significant difference in resistant strain with and without drug for nitrite concentration and CFU count (Fig. 2e, f: Mann-Whitney U-Test: U = 33.5, *p*-value 0.57 for nitrite and U = 40.5 and *p* = 1.0 for CFU count).

**Nitrite in patients with or without confirmed urinary tract infection**
Samples from patients with UTI has higher nitrite concentration than non-UTI samples, with a Mann-Whitey U-test revealing significant difference between non UTI and UTI patients (Mann-Whitey U-test U = 102.5 and *p*-value < 0.001). Boxplots show the distribution of creatinine-corrected nitrite across categories from the 25th to the 75th percentile, with a line inside the box indicating the median. (Fig. 3a). Spearman correlation between bacterial count and generated nitrite (corrected for renal clearance using detected creatinine levels) reveals a significant relationship (Spearman correlation: 0.46 and *p*-value 0.032) (Fig. 3b) when excluding false low nitrite levels (nitrite <1uM and CFU/mL >0.2 × 10⁶, red crosses). The correlation between absolute nitrite levels and CFUs is not significant (Supplementary Fig. 1).

## Discussion
In an in-vitro model, we show that nitrite produced by *E. coli* is significantly correlated with CFU count in all tested strains. A Mann-Whitey U-Test shows that nitrite concentration and CFU count are significantly different in the susceptible strain with and without antimicrobial drugs, while no significant difference is observed in the resistant strain with and without drug. Furthermore, our study reveals that concentrations of antimicrobial drugs, effective in inhibiting bacterial growth, prevent the reduction of nitrates to nitrite in artificial urine. Spearman correlation analysis shows a significant correlation between CFU count and generated nitrite in human urine samples.

These findings confirm a relationship between CFU count and nitrite production both in vitro and in human urine, suggesting that longitudinal monitoring of nitrite could serve as a valuable indicator of UTI progression and potentially the efficacy of drug therapy.

At low oxygen concentrations, *E. coli* utilises nitrate as a secondary electron acceptor for anaerobic respiration[15], making nitrite a commonly used indicator for UTIs. This has limitations, such as no increase in nitrite if no dietary nitrate is present in urine, or infection with non-nitrate reducing strains.

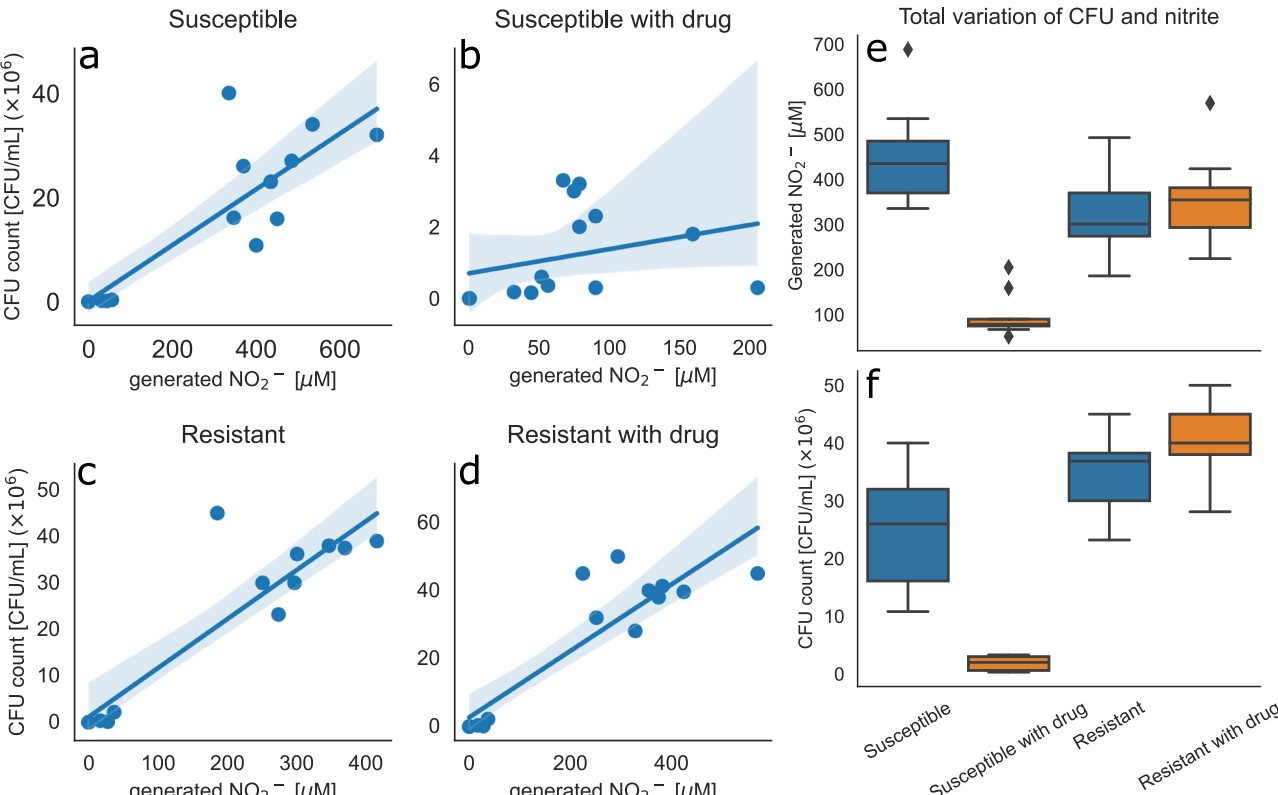

**Fig. 2 | Correlation of bacterial growth and generated nitrite in in-vitro model and comparison between susceptible and resistant *E. coli* strain.** CFU count is correlated with nitrite production of each individual sample. The Spearman correlation of CFU count and nitrite for all strains namely susceptible strain **a)** (Spearman correlation: 0.82, *p*-value < 0.001), susceptible strain with drug **b)** (Spearman correlation: 0.62, *p*-value: 0.014), the resistant strain **c)** (Spearman correlation: 0.86, *p*-value < 0.0001) and the resistant strain with antimicrobial drug **d)** (Spearman correlation: 0.79, *p*-value < 0.001) are significant. The solid line in **a-d** represents the linear least square fit, and the shaded area is the 95% confidence limits. Boxplots show the distribution of CFU counts (**f**) and generated nitrite (**e**) across categories from the 25th to the 75th percentile, with a line inside the box indicating the median. The whiskers extend from the box to the smallest and largest values within 1.5 times the interquartile range. Data points outside this range are considered outliers and are shown as individual points (lozenges). Total generated nitrite (**e**) and CFU count (**f**) after the addition of antimicrobial drug (time-points 3-5) show a significant difference between the drug-treated and untreated conditions in the susceptible strain. In contrast, no significant difference is observed in the resistant strains, regardless of drug treatment.

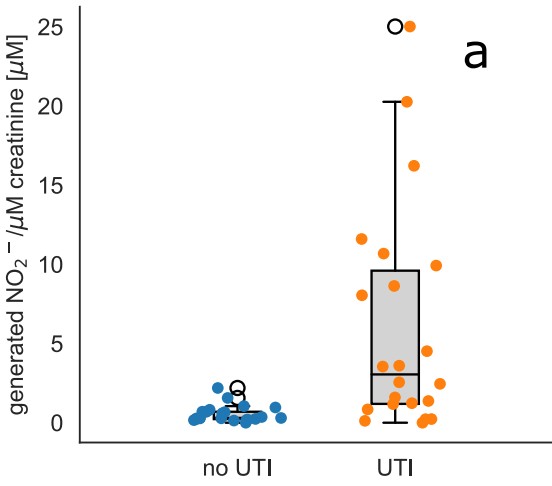

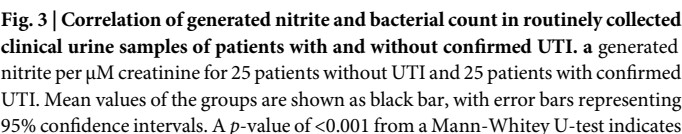

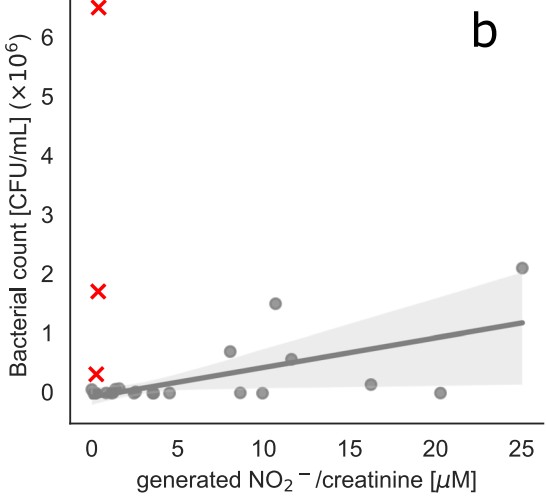

**Fig. 3 | Correlation of generated nitrite and bacterial count in routinely collected clinical urine samples of patients with and without confirmed UTI. a** generated nitrite per μM creatinine for 25 patients without UTI and 25 patients with confirmed UTI. Mean values of the groups are shown as black bar, with error bars representing 95% confidence intervals. A *p*-value of <0.001 from a Mann-Whitey U-test indicates statistical significant difference between the groups. **b** generated nitrite per μM creatinine for UTI patients is correlated with bacterial count (Spearman correlation: 0.46, *p*-value: 0.0032), when 3 UTI samples tested false negative for nitrite (red) were excluded. The solid line in **b**) represents the linear least squares fit, and the shaded area indicates the 95% confidence interval around the fit.

In this study, nitrite and CFU in 25 urine samples from patients with confirmed *E. coli* UTI were compared to 25 non-UTI patient samples. Five UTI samples demonstrate no growth when re-plated for quantitative CFU count. This may be explained by the 4 days delay between sample collection and testing or strains incompatible to grow on the used plates. Three UTI samples show low generated nitrite (<1 uM) but high bacterial count ($>0.2 \times 10^6$ CFU/mL), which could be explained by no nitrate present in urine so no conversion to nitrite is possible. The low nitrite with high CFU results observed in three patient samples highlight a limitation of nitrite-based monitoring, particularly in cases where nitrate is not present in the urine, or in infections with non-nitrate reducing bacteria.

While significant, the correlation between CFU and nitrite in patient samples is weak, which might be due to sample storage, heterogenous infecting organisms, and variations of pH. The reported sensitivities for nitrite as a biomarker for UTI infection are mixed[4] and should be interpreted alongside other biomarkers on dipstick tests according to the National Institute for Health and Care Excellence protocol[16]. Additionally, future research should explore methods that enable accounting for the variability in nitrate levels among patients, such as analysing nitrate alongside nitrite to exclude nitrate-negative samples and incorporate a range of antimicrobial drugs to assess whether the drug itself influences biomarker formation[17].

Our findings demonstrate that even biomarkers with limitations could provide valuable pharmacodynamic insights when monitored over time. Dipstick tests are accessible, affordable, and have been integrated into the clinical workflow for testing UTIs in GP practices and hospitals and are used for home-testing[16]. Rather than using urine biomarkers solely for UTI identification, they could be leveraged to estimate PD and track infection progression. In a future study, urine samples from a diverse cohort of patients should be collected longitudinally through patient follow-ups over the course of a range of antimicrobial treatments. Novel and more promising host-response biomarkers such as Xanthine Oxidase and Myeloperoxidase, which have shown high sensitivity and selectivity in identifying UTIs, could then be explored as marker of PD and antimicrobial efficacy, alongside nitrite and other frequently deployed biomarkers tested with urinary dipstick tests.

Our findings suggest the possibility of using longitudinal nitrite monitoring as an in vivo measure of antimicrobial PD in the context of UTI, which should be explored for other UTI biomarkers. Future research could focus on validating nitrite monitoring in larger, more diverse patient populations, and exploring its potential to be used alongside host-response biomarkers such as leukocytes to simultaneously track and respond to the patients' immune reaction against the infecting organism. The integration of quantitative nitrite monitoring, alongside host-response biomarker and drug levels into point-of-care diagnostic technologies could improve UTI treatment by enabling real-time, non-invasive assessment of infection progression and antimicrobial efficacy.

## Data availability

Source data is available in Figshare with the identifier https://doi.org/10.6084/m9.figshare.28733081.v2[18].

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

## Acknowledgements

Ellen Stadler is funded by the President's PhD Scholarship at Imperial College London. The research was partially funded by the National Institute for Health Research Health Protection Research Unit (NIHR HPRU) in Healthcare Associated Infections and Antimicrobial Resistance at Imperial College London in partnership with the UK Health Security Agency (previously PHE) in collaboration with Imperial Healthcare Partners, University of Cambridge and University of Warwick. The views expressed in this publication are those of the author(s) and not necessarily those of the NHS, the National Institute for Health Research, the Department of Health and Social Care or the UK Health Security Agency. Professor Alison Holmes is a National Institute for Health Research (NIHR) Senior Investigator. The views expressed in this publication are those of the author(s) and not necessarily those of the Department of Health and Social Care NHS, or the National Institute for Health Research.

## Author contributions

Conceptualisation: E.V.S., D.O., T.M.R. Investigation: E.V.S., Visualization: E.V.S., D.O., T.M.R. Funding acquisition: E.V.S., A.H., D.O., T.M.R. Supervision: A.H., D.O., T.M.R., Writing—original draft: E.V.S., D.O., T.R., Writing—review & editing: E.V.S., A.H., D.O., M.S., C.B., T.M.R.

## Competing interests

The authors compare the following competing interests: Timothy Miles Rawson is an Editorial Board Member for Communications Medicine, but was not involved in the editorial review or peer review, nor in the decision to publish this article. All other authors declare no competing interests.

## Additional information

[1]Centre for Antimicrobial Optimisation, Imperial College London, London, UK. [2]Department of Infectious Diseases, Imperial College London, London, UK. [3]National Institute for Health Research, Health Protection Research Unit in Healthcare Associated Infections and Antimicrobial Resistance, Imperial College London, London, UK. [4]David Price Evans Infectious Diseases & Global Health Group, The University of Liverpool, Liverpool, UK. [5]Fleming Institute, Imperial College London, London, UK. [6]Department of Bioengineering, Imperial College London, London, UK. [7]Antimicrobial Discovery, Development and Diagnostics (AD3) UK Health Security Agency, Porton Down, Salisbury, Wiltshire, UK. [8]Institute of Pharmaceutical Science, King's College London, London, UK. [9]Healthcare Associated Infections, Fungal, Antimicrobial Resistance, Antimicrobial Use, and Sepsis Division, UK Health Security Agency, London, UK. ✉e-mail: ellen.stadler20@imperial.ac.uk

