## [Peer Review file · Communications Medicine]

Feasibility of serial measurement of nitrite for pharmacodynamic monitoring and precision prescribing in urinary tract infections

Corresponding Author: Ms Ellen Stadler

Version 0:

Reviewer comments:

Reviewer #1

(Remarks to the Author)

Stadler et al. present a study where they show, that nitrite concentration in urine correlates with CFUs of E. coli in artificial urine in vitro, and that antibiotic inhibition of bacterial growth halts the nitrite production. Therefore, measurement of nitrite concentration in urine could be used as a marker of effect or no effect of antibiotics for treatment of urinary tract infection. The work is nicely performed and presented.

The variation of nitrite concentration in urine samples from patients with E.coli UTI shown in Figure 3 reveals some of the problems with trying to quantify nitrite as related to E.coli presence in the urine. E.coli produces both a nitrate reductase AND a nitrite reductase, as do other Enterobacterales and other species. The reduction of nitrate in urine by E.coli depends on a number of factors e.g. the duration of the reaction between E.coli and nitrate, pH, temperature and others. The same counts for nitrite reductase. This is probably the reason for the large variation in nitrite concentration in vivo (in the patient urines). This is also the reason for why the dipstick test with nitrite an leucocyte reductase shows variable results even with $>10^5$ E.coli in urine. The measurement of nitrite in urine could probably be used as a test for antibiotic activity in the urine, but the sensitivity and specificity need to be studied in real life - and should be compared with the simple quantitative counts of E.coli in urine as a comparator for antibiotic activity.

Specific comment:

1. In Figure 3 the nitrite values should be shown as medians and percentiles; they do not look normally distributed and the authors also use the Mann-Whitney test. In figure B, a correlation coefficient of 0.46, although significant, means that around 21% of the variation in data points is explained by variation in X and Y factors - and that is after removing the outliers.

2. One wonders what boric acid does to the growth of bacteria and nitrate and nitrite. The authors should try to perform the same experiments on patient samples without boric acid and refrigerated immediately after sampling.

Reviewer #2

(Remarks to the Author)

The manuscript presents a well-structured and compelling study with a strong experimental design. However, further clarification on statistical analysis, reproducibility, and potential limitations would enhance its impact.

I want to add a few comments on the following topics: novelty, influence on the field, methodology, and limitations.

Novelty & Significance: The study addresses an underexplored area, but comparisons to other biomarkers in UTI diagnostics would strengthen its positioning.

Strength of Conclusions: Correlation does not imply causation—would in vivo studies or patient follow-ups reinforce these findings? Also, how do dietary nitrate variations affect nitrite levels?

Influence on the Field: This work has clinical potential but requires validation in larger, diverse cohorts.

Reproducibility & Methodology: While detailed, additional clarity on bacterial growth variations and batch effects in nitrite measurement would improve reproducibility.

Statistical Analysis: Using Spearman correlation and Mann-Whitney U-tests is appropriate but would benefit from further justification. Were corrections for multiple comparisons applied?

Limitations: The authors acknowledge dietary nitrate variability and non-nitrate-reducing bacteria as concerns but should discuss mitigation strategies for potential false negatives.

There were also some grammatical errors (e.g., "did not effect growth" → "did not affect growth"). The authors should define acronyms upon first mention for clarity.

This study provides promising insights into nitrite as a UTI biomarker but would benefit from further validation, methodological clarifications, and statistical robustness.

Reviewer #3

(Remarks to the Author)

This study describes monitoring nitrite concentrations as a biomarker in the management of UTIs. The paper is nicely done and will be of interest to clinicians and researchers. My specific comments and suggestions are as follows:

- 1) The methods are sufficiently detailed and should allow for replication by other investigators.
- 2) There should be a statistics section in the methods, especially since the authors used P values and Spearman correlation.
- 3) In Figure 2, (Mann-Whitney U test).
- 4) Do the authors think amoxicillin could have influenced nitrate to nitrite reduction compared to a different antibiotic? See Li Z et al. Selective stress of antibiotics on microbial denitrification: Inhibitory effects, dynamics of microbial community structure and function. J Hazard Mater 2021;405:124366.

Version 1:

Reviewer comments:

Reviewer #1

(Remarks to the Author)

The authors have responded adequately to the comments from the reviewers. I have no further comments.

Reviewer #2

(Remarks to the Author)

Thank you for addressing our previous review comments and for the improvements made to the manuscript. The revisions have significantly enhanced the clarity and scientific rigor of the study. Below are some additional remarks and minor areas that could further strengthen the manuscript:

Comparison to Other UTI Diagnostic Methods:

While the discussion includes nitrite's potential as a biomarker, providing a more detailed comparison with other UTI diagnostic tools (e.g., leukocyte esterase, dipstick tests, culture-based methods) would be beneficial. This would help contextualize the advantages and limitations of nitrite monitoring within current clinical practice.

Multiple Comparisons in Statistical Analysis:

The manuscript effectively explains the choice of statistical tests; however, it does not explicitly state whether corrections for multiple comparisons were applied. If multiple tests were performed, a brief mention of any correction method used (e.g., Bonferroni correction) would enhance statistical rigor.

Clinical Implementation Considerations:

The study suggests that longitudinal nitrite monitoring could be useful for antimicrobial pharmacodynamics. While this is a promising concept, a brief discussion on how this could be integrated into clinical workflows (e.g., bedside testing, point-of-care diagnostics) would help clarify its practical applications.

Clarification of Statistical Methods:

Including a dedicated statistics section in the methods would improve transparency, especially given the use of p-values and Spearman correlation.

Effect of Amoxicillin on Nitrate Reduction:

It would be useful to discuss whether amoxicillin specifically influences nitrate to nitrite reduction, compared to other antibiotics, as suggested in the literature (Li Z et al., 2021).

Boric Acid Impact on Urine Samples:

Consider addressing whether boric acid affects bacterial growth and nitrite/nitrate levels, as Reviewer #1 pointed out. If feasible, an experiment without boric acid could strengthen future studies.

Overall, the revisions have greatly improved the manuscript, and addressing these minor points would further strengthen its impact. Thank you for your hard work!!

Reviewer #3

(Remarks to the Author)

The authors have done an appropriate job in addressing my and the other reviewer's comments. No further changes are necessary in my opinion.

Version 2:

Reviewer comments:

Reviewer #2

(Remarks to the Author)

The authors have responded adequately to the comments. I have no further changes.

Reviewer 1

Reviewer comment	Response	Action
Stadler et al. present a study where they show, that nitrite concentration in urine correlates with CFUs of E. coli in artificial urine in vitro, and that antibiotic inhibition of bacterial growth halts the nitrite production. Therefore, measurement of nitrite concentration in urine could be used as a marker of effect or no effect of antibiotics for treatment of urinary tract infection. The work is nicely performed and presented. The variation of nitrite concentration in urine samples from patients with E.coli UTI shown in Figure 3 reveals some of the problems with trying to quantify nitrite as related to E.coli presence in the urine. E.coli produces both a nitrate reductase AND a nitrite reductase, as do other Enterobacteriales and other species. The reduction of nitrate in urine by E.coli depends on a number of factors e.g. the duration of the reaction between E.coli and nitrate, pH, temperature and others. The same counts for nitrite reductase. This is probably the reason for the large variation in nitrite concentration in vivo (in the patient urines). This is also the reason for why the	The authors agree with Reviewer 1's comments. The paper addresses the limitations of nitrite as a bacteria-specific biomarker, both in general and for estimating antimicrobial killing - such as when dietary nitrate is absent in the urine or when the infection is caused by non-nitrate-reducing bacteria. We used nitrite to demonstrate that even a biomarker with limitations has the potential to provide insights into pharmacodynamics (PD) in UTIs when its concentration or even a simple yes-or-no result is monitored over time. Our goal was to highlight that urine biomarkers can be used not only for UTI identification but also for tracking PD and infection progression. To clarify this, we expanded the discussion, emphasizing that this approach should be explored further for novel and more promising biomarkers, such as Xanthine Oxidase and Myeloperoxidase, which have shown high sensitivity and selectivity in identifying UTIs.	Added lines 187-201

dipstick test with nitrite an leucocyte reductase shows variable results even with $>10^5$ E.coli in urine. The measurement of nitrite in urine could probably be used as a test for antibiotic activity in the urine, but the sensitivity and specificity need to be studied in real life - and should be compared with the simple quatitative counts of E.coli in urine as a comparator for antibiotic activity.		
In Figure 3 the nitrite values should be shown as medians and percentiles; they do not look normally distributed and the authors also use the Mann-Whitney test.	Figure 3 A has been adapted to show medians and percentiles and description in text has been updated accordingly. Additionally, Figure 5 in appendix has been adjusted accordingly.	Amended figure 3 line 163 and Appendix figure 5 line 267 Added line 156-157
In figure B, a correlation coefficient of 0.46, although significant, means that around 21% of the variation in data points is explained by variation in X and Y factors - and that is after removing the outliers.	Thank you for your comment. We acknowledge that a correlation coefficient of 0.46, while significant, suggests a weak relationship between the variables. Nitrite, as a bacteria-specific biomarker for UTIs, has significant limitations. These include false-negative results due to the absence of dietary nitrate in urine or infections caused by non-nitrate-reducing bacteria. Sensitivity and selectivity are often low in various studies. Despite these flaws, nitrite remains a biomarker due to the rapid and cost-effective dipstick tests, which utilize the colorimetric	Added lines 187-201.

	Griess reaction. Nitrite and other biomarkers like leucocyte esterase are recommended for guiding treatment decisions in the National Institute for Health and Care Excellence (NICE) protocol by Public Health England [1]. Regarding the in-vivo results section, we used routinely collected urine samples tested for bacterial growth and nitrite concentration 4 days after collection and storage. In the used ethics protocol (IRAS project ID: 162013, in methods), only routinely collected clinical samples can be used – fresh samples, therefore, were not an option for us. The weak correlation between CFU counts and nitrite generation in vitro, compared to in vivo results, may be explained by heterogeneous infecting organisms, inhibition of bacterial growth due to storage, and variations in pH, which has been added as a comment in the discussion. As stated in the original draft and further emphasized in the discussion, our study serves as a proof-of-concept, showing that monitoring biomarkers quantitatively over time could provide valuable insights into treatment efficacy and antimicrobial resistance, and other	
--	---	--

	biomarkers should be explored similarly in future studies.	
One wonders what boric acid does to the growth of bacteria and nitrate and nitrite. The authors should try to perform the same experiments on patient samples without boric acid and refrigerated immediately after sampling.	Considering the impact of boric acid on bacterial growth and nitrate/nitrite levels in urine is crucial in our study, as it directly influences our patient sample results. Boric acid is widely used in the NHS for its bacteriostatic properties. At the concentrations found in commercially available boric acid urine containers, boric acid is non-toxic and effectively preserves bacterial counts as they were at the time of sampling [2]. Boric acid has no impact on leucocyte esterase or nitrite detection with dipsticks [3]. Since our study uses the same chemical reaction (the Griess test) as dipstick nitrite detection, we can be confident that nitrite readings remain unaffected. Under our current ethics protocol, we are restricted to using routinely collected urine samples, which are stored with boric acid. In a planned future study, we will collect fresh urine samples directly from patients, which will be conducted under a new ethics protocol.	Added lines 192 - 201

Reviewer comment	Response	Action
The manuscript presents a well-structured and compelling study with a strong experimental design. However, further clarification on statistical analysis, reproducibility, and potential limitations would enhance its impact. I want to add a few comments on the following topics: novelty, influence on the field, methodology, and limitations. Novelty & Significance: The study addresses an underexplored area, but comparisons to other biomarkers in UTI diagnostics would strengthen its positioning.	Thank you for your insightful comment. We agree that expanding the scope to include additional biomarkers would further strengthen the potential of UTI biomarkers for monitoring infection progression and treatment efficacy. Our in vitro study focused on bacteria-specific biomarkers, as host-response markers like leukocyte esterase would not be elevated in this controlled setting. However, we have referenced studies showing that host-response biomarkers, such as leukocyte esterase, decrease in women with UTIs after successful treatment but remain elevated after failed treatment [4]. As addressed in your comment below, we recognize the importance of studying larger and more diverse patient cohorts with longitudinal follow-ups. Future research should include a broader range of biomarkers, particularly promising candidates like Xanthine Oxidase and Myeloperoxidase, to enhance diagnostic and monitoring capabilities. A statistical analysis section and methods were amended as	

	proposed in your comments below.	
Strength of Conclusions: Correlation does not imply causation—would in vivo studies or patient follow-ups reinforce these findings? Also, how do dietary nitrate variations affect nitrite levels? Influence on the Field: This work has clinical potential but requires validation in larger, diverse cohorts.	Thank you for your comment. We agree that patient follow-ups are essential to reinforce our findings. However, in a proof-of-concept setting, this is challenging since urine samples are typically submitted only once at a GP visit. To address this limitation, we are working on establishing a more controlled study that will enable longitudinal monitoring. This study will also incorporate additional host-response biomarkers alongside novel and promising UTI biomarkers such as Xanthine Oxidase and Myeloperoxidase.	Added lines 196-201
Reproducibility & Methodology: While detailed, additional clarity on bacterial growth variations and batch effects in nitrite measurement would improve reproducibility.	The section “growth conditions and antimicrobial killing” was amended for clarity.	Added lines 77-79, lines 81-84, lines 86-88
Statistical Analysis: Using Spearman correlation and Mann-Whitney U-tests is appropriate but would benefit from further justification. Were corrections for multiple comparisons applied?	Spearman correlation (Figure 2 A–D) and Mann-Whitney U-tests (Figure 2 E–F) were performed on separate data subsets, and thus multiple comparison correction was not required. Further justification was added in a short statistical analysis method section, as proposed by reviewer 3.	Added lines 122-134

Limitations: The authors acknowledge dietary nitrate variability and non-nitrate-reducing bacteria as concerns but should discuss mitigation strategies for potential false negatives.	We agree that UTI infections caused by non-nitrate-reducing bacteria, as well as cases where there is little or no dietary nitrate in the urine, present challenges not only for using nitrite as a UTI biomarker but also for its role in monitoring treatment success over time. In the discussion section, we propose that nitrite be used in conjunction with host-response biomarkers and other novel, promising biomarkers to enhance its diagnostic value and mitigate these limitations, like how leucocyte esterase is currently used alongside nitrite in dipstick tests.	Added lines 187-191
There were also some grammatical errors (e.g., "did not effect growth" → "did not affect growth"). The authors should define acronyms upon first mention for clarity.	Grammatical errors have been addressed, and the document has been checked so all acronyms are defined upon the first time mentioned.	Corrected line 138
This study provides promising insights into nitrite as a UTI biomarker but would benefit from further validation, methodological clarifications, and statistical robustness.	Thank you for your comment. We addressed comments from all reviewers and acknowledges limitations where appropriate. This study lays the groundwork for a future study looking at patients longitudinally, which has been added to the discussion.	

Reviewer 3

Reviewer comment	Response	Action
This study describes monitoring nitrite concentrations as a	Thank you for your comment. We appreciate your input and have	

biomarker in the management of UTIs. The paper is nicely done and will be of interest to clinicians and researchers. My specific comments and suggestions are as follows:	provided point by point responses to the comments below.	
The methods are sufficiently detailed and should allow for replication by other investigators.	Thank you for your comment. According to feedback of reviewer 2, we have added a few comments to the materials and methods section to make the growth conditions clearer.	Added lines 77-79, lines 81-84, lines 86-88
There should be a statistics section in the methods, especially since the authors used P values and Spearman correlation.	Statistical analysis section was included.	Added lines 122-134
In Figure 2, (Mann-Whitney U test).	The typo in the figure legend has been corrected.	Amended figure description in line 152
Do the authors think amoxicillin could have influenced nitrate to nitrite reduction compared to a different antibiotic? See Li Z et al. Selective stress of antibiotics on microbial denitrification: Inhibitory effects, dynamics of microbial community structure and function. J Hazard Mater 2021;405:124366. reference in paper	Thank you for the comment, we agree it is important to consider the potential influence of antibiotics like amoxicillin on the nitrate to nitrite reduction process. As noted by Li Z et al. (2021), the selective stress of antibiotics on microbials has effects on their denitrification capabilities. In the future, we aim to investigate nitrite alongside host-response biomarkers longitudinally through patient follow ups for a more robust PD estimation, which has been added at the end of the discussion section.	Added lines 196-201

Reviewer 1

Reviewer comment	Response	Action
The authors have responded adequately to the comments from the reviewers. I have no further comments.	Thank you for the valuable feedback and we are glad we could address comments adequately.	

Reviewer 2

Reviewer comment	Response	Action
Thank you for addressing our previous review comments and for the improvements made to the manuscript. The revisions have significantly enhanced the clarity and scientific rigor of the study. Below are some additional remarks and minor areas that could further strengthen the manuscript:	The authors would like to thank Reviewer 2 for the constructive and helpful feedback and comments. We are glad some could be addressed in previous revision.	
Comparison to other UTI Diagnostic Methods: While the discussion includes nitrite's potential as a biomarker, providing a more detailed comparison with other UTI diagnostic tools (e.g., leukocyte esterase, dipstick tests, culture-based methods) would be beneficial. This would help contextualize the advantages and limitations of nitrite monitoring within current clinical practice.	Thank you for your insightful comment. We agree other UTI biomarkers such as leukocyte esterases and biomarkers described more recently for UTI diagnosis such as Xanthine Oxidase and Myeloperoxidase should be explored as marker of PD and antimicrobial efficacy in UTIs. We would like to note that our paper focuses on point-of-care technologies that are rapid and do not require specialised equipment or a laboratory, which could be performed by patients at home or at GP practices. We therefore clarified that in future, longitudinal in-human studies should include a variety of results such as leukocyte esterase, protein, all tested using dipstick tests.	The final paragraph from 196-201 has been amended to clarify that in future, longitudinal in-human studies, other biomarkers tested with dipstick tests should be explored alongside nitrite: Novel and more promising host-response biomarkers such as Xanthine Oxidase and Myeloperoxidase, which have shown high sensitivity and selectivity in identifying UTIs, could then be explored as marker of PD and antimicrobial efficacy, alongside nitrite and other frequently deployed biomarkers tested with urinary dipstick tests.

Multiple Comparisons in Statistical Analysis: The manuscript effectively explains the choice of statistical tests; however, it does not explicitly state whether corrections for multiple comparisons were applied. If multiple tests were performed, a brief mention of any correction method used (e.g., Bonferroni correction) would enhance statistical rigor.	Thank you for your comment. Spearman correlation (Figure 2 A–D) and Mann-Whitney U-tests (Figure 2 E–F) were performed on separate data subsets, and thus multiple comparison correction was not required. We clarified the statistical approach by adding a dedicated statistics section, as suggested in your following comment.	A statistical analysis section was added, lines 122-134
Clarification of Statistical Methods: Including a dedicated statistics section in the methods would improve transparency, especially given the use of p-values and Spearman correlation.	We agree with Reviewer 2's comment that a statistical analysis section would improve transparency. A statistical section was included in the revised manuscript.	Added lines 122-134
Clinical Implementation Considerations: The study suggests that longitudinal nitrite monitoring could be useful for antimicrobial pharmacodynamics. While this is a promising concept, a brief discussion on how this could be integrated into clinical workflows (e.g., bedside testing, point-of-care diagnostics) would help clarify its practical applications.	Thank you for this constructive feedback. Urinary dipstick tests are widely used due to their accessibility and affordability, and their use has been integrated into GP and hospital workflows. We agree that mentioning their integration into the clinical workflow, as well as their potential for use by GPs and patients to track UTIs at home, would add value to the proposal.	Added lines 198-200: Dipstick tests are accessible, affordable, and have been integrated into the clinical workflow for testing UTIs in GP practices and hospitals and are used for home-testing.
Effect of Amoxicillin on Nitrate Reduction: It would be useful to discuss whether amoxicillin specifically influences nitrate to nitrite reduction, compared to other antibiotics, as suggested in the literature (Li Z et al., 2021).	Thank you for the comment, we agree it is important to consider the potential influence of antibiotics like amoxicillin on the nitrate to nitrite reduction process. In the future, we aim to investigate nitrite alongside host-response biomarkers longitudinally through patient follow ups for a more robust PD estimation, which has been added at the end of the discussion section. Additionally, we added a comment on exploring the influence of the antimicrobial	Added lines 195: [...] and incorporate a range of antimicrobial drugs to assess whether the drug itself influences biomarker formation. Added the relevant reference proposed by reviewer 2.

	drug itself on the biomarker and added the relevant reference.	
Boric Acid Impact on Urine Samples: Consider addressing whether boric acid affects bacterial growth and nitrite/nitrate levels, as Reviewer #1 pointed out. If feasible, an experiment without boric acid could strengthen future studies.	The impact of boric acid on bacterial growth and nitrate/nitrite levels in urine is an important factor in our study, as it directly affects the results from patient samples. Boric acid is commonly used in the NHS for its bacteriostatic properties. At the concentrations present in commercially available boric acid urine containers, it is non-toxic and effectively preserves bacterial counts as they were at the time of sampling (Hedström et al., 2021). Boric acid does not interfere with leucocyte esterase or nitrite detection using dipsticks (Raff & Bazzetta, n.d.). Since our study uses the same chemical reaction (the Griess test) for nitrite detection as the dipstick method, we can be confident that nitrite readings will remain unaffected. A relevant reference has been added in the methods section. In accordance with our current ethics protocol, we are restricted to using routinely collected urine samples, which are stored with boric acid. In a future planned study, we will collect fresh urine samples directly from patients, which will be conducted under a new ethics protocol.	Added reference: Raff, L. J., & Bazzetta, K. (n.d.). Leukocyte Esterase and Nitrite Testing of Urine Preserved with Boric Acid. And added lines 114-115: Boric acid preserves bacterial counts and has no known effect on nitrite detection using dipstick tests.
Overall, the revisions have greatly improved the manuscript, and addressing these minor points would further strengthen its impact. Thank you for your hard work!!	The authors would like to thank Reviewer 2 for their insightful and valuable comments. We are pleased to have had the opportunity to address them.	

Reviewer 3

Reviewer comment	Response	Action
The authors have done an appropriate job in addressing my and the other reviewer's comments. No further changes are necessary in my opinion.	We would like to thank reviewer 3 for the constructive feedback and comments and are glad we were able to address them.